# Reading and Misleading: Changes in Head and Eye Movements Reveal Attentional Orienting in a Social Context

**Tom Foulsham \*, Monika Gejdosova and Laura Caunt**

Department of Psychology, University of Essex, Colchester, CO4 3SQ, UK
* **\*** Correspondence: foulsham@essex.ac.uk

**Abstract:** Social attention describes how observers orient to social information and exhibit behaviors such as gaze following. These behaviors are examples of how attentional orienting may differ when in the presence of other people, although they have typically been studied without actual social presence. In the present study we ask whether orienting, as measured by head and eye movements, will change when participants are trying to mislead or hide their attention from a bystander. In two experiments, observers performed a preference task while being video-recorded, and subsequent participants were asked to guess the response of the participant based on a video of the head and upper body. In a second condition, observers were told to try to mislead the "guesser". The results showed that participants' preference responses could be guessed from videos of the head and, critically, that participants spontaneously changed their orienting behavior in order to mislead by reducing the rate at which they made large head movements. Masking the eyes with sunglasses suggested that head movements were most important in our setup. This indicates that head and eye movements can be used flexibly according to the socio-communicative context.

**Keywords:** social attention; gaze; head movements

## 1. Introduction

When humans pay attention, they frequently do so in the presence of other people. In other words, the processes that we use to prioritize certain information are often displayed within a social group. Research into "social" attention has yielded a number of different insights [1,2]. In brief, such research can be divided into two sorts of findings. First, it has been observed that humans prioritize social information, and in particular images of other people, by spontaneously selecting them with covert and overt attention [3–5]. This selection of social stimuli appears to happen quickly in a range of simple and complex paradigms and even when it would be beneficial to not select this information [6]. Hence, it has been argued that images of people are special and may trigger automatic capture of visual attention in laboratory tasks. The degree to which this spontaneous attentional bias is observed may vary considerably, however, with recent research finding no difference in manual reaction time measures when stimuli are carefully controlled [7].

Second, attention, along with many other aspects of cognition and behavior, changes when one is in the presence of other people [8]. Observers frequently follow the gaze of other pedestrians, and the way that this change in visual attention spreads through crowds can be modelled as a dynamic process [9]. However, the effect of other people on our attention depends a lot on the context. For example, participants may avert their gaze when in a social situation [10], or they may choose to look at different things when there is a real or implied social presence [11].

When these two sets of findings are considered together, it is perhaps not surprising that a number of recent studies have found discrepancies between the way that participants respond to images of people and the way that they respond to people who are physically present [2,12]. For example, while it is established that people will spontaneously move their overt and covert attention to images of faces, they are less likely to gaze at a real person [13]. Gaze following in real situations is also less common than one might expect from laboratory experiments and is less likely when pedestrians are walking towards the observer [9,14]. One reason for differences in social attention to images and real people is that eye gaze has a dual function: It extracts information from the environment but it also provides a communicative signal to other observers [2,15]. This signaling function of overt visual attention is likely to be both rich and nuanced, as indicated by research into gaze patterns in conversation, where conversants make and break gaze regularly as part of the pattern of turn taking in speech [16]. The present study extends a novel paradigm which allows researchers to study the signaling function of attentional orienting.

Foulsham and Lock [17] recorded participants' eye movements while they completed a simple preference task which required choosing which of four images on a screen was most preferred. They then replayed an animation of these eye movements to the next participant and asked them to guess which item had been chosen. Thus, even though the stimuli in this experiment were not biologically or social relevant, the guessing participants were required to read social signals from a representation of overt attention. The results showed that participants were able to do this relatively well, revealing how people can use gaze following to draw inferences about what someone else is doing or thinking. Foulsham and Lock then asked the same participants to repeat the preference task, but this time to do so while trying to mislead the person who would be watching their gaze movements. Being able to deceive others in this case requires a sophisticated understanding of how they will read a situation (and therefore, arguably, a theory of mind; [18]). If attentional orienting is only involved in extracting information in a relatively automatic way, then it might not differ when the communicative context is changed in order to require deception. In contrast, if participants can spontaneously change where they look in order to deceive, it shows that participants know the signals that are being provided and can modify them in a sophisticated way. The results of Foulsham and Lock confirmed this latter hypothesis. Participants changed their overt attention by looking less at the chosen item, and this resulted in a decline in guessing performance in the "lying" trials.

As reviewed above, the "eyes-lies" paradigm described by Foulsham and Lock [17] showed how eye movements to non-social stimuli may differ depending on the socio-communicative context. However, in that study the observers were required to "read" another person's attention in response to a symbolic representation of eye fixations (an animation of a moving dot). This is clearly very different to the type of social information that is available to us in real, face-to-face situations. In real situations we instead must respond to cues from eye, head and body positions which indicate where somebody is paying attention. Critically, the degree to which these different signals can be read is likely to vary. For example, while quick eye movements may be quite subtle, a large head movement is likely to be more obvious. It is well established that gaze in the real world involves a coordination of body, head and eye movements [19], and in some cases we may also have a choice as to whether we attend to something overtly or covertly (i.e., without any change in gaze at all). Very little is known about how flexible these systems of attention are, and whether they can be influenced by social context.

In the present study we asked whether participants can reliably infer attention and preference from videos of the head, just as they can from a symbolic representation of eye movements [17]. If so, we can then ask which signals are being displayed, and whether they can be spontaneously changed in order to communicate or mislead another person. There are different ways in which people might change their orienting behavior to deceive. For example, they might look away from something which they are choosing, as they did in our previous study [17]. However, they might also change their looking behavior in a more general way, by reducing the number of overt shifts of attention that they produce. If this is the case, then we would expect to see a reduction in the number of fixations and/or head movements.

## 2. Experiment 1

In Experiment 1, we repeated the general procedure from Foulsham and Lock [17], but with a recording setup which allowed us to capture head and eye movements while participants were performing a preference task on a large screen.

*Materials and Methods*

**Participants.** Twenty student volunteers from the University of Essex participated in the experiment (13 females; mean age = 21.8 years). This sample size was chosen based on our previous study [17], which resulted in a large effect size for the difference between Truth and Lie trials. Sample size calculations indicated that in a within-subjects design a sample size of 10 was sufficient for detecting this effect with 95% power. All participants had normal or corrected vision. An additional pilot participant was tested due to the requirement for participants to react to another person's behavior. The pilot participant provided preference responses for the first experimental participant. Head movement and fixation data of the pilot participant were not included in the analysis.

All participants gave their informed consent. The study was conducted in accordance with the Declaration of Helsinki, and the protocol was approved by the University of Essex Ethics Committee (project ID code: TF150).

**Stimuli and apparatus.** Figure 1 gives an overview of the method and materials. In the first and the last part of the experiment, participants made a choice each trial from 4 computer-generated patterns (fractals). All images were colored fractals, sourced from freely available collections online (and were the same as those used in Foulsham & Lock [17]). One hundred and forty four fractals were used in total. For each trial, the fractals were allocated at random to a group of four and presented in 2 × 2 matrix on a black background.

Stimuli were displayed on a large screen which measured 150 × 110 cm, projected via a rear projection projector. Participants were asked to stand on a spot 60 cm from the screen. The room was dimly lit and lighting was controlled by a separate lamp behind the participant, to prevent reflection but allow enough ambient light for webcam recordings. Presentation and recording during the preference task were controlled by an iMac and custom Python code. Participants entered their preference response using a Microsoft Sidewinder gaming controller connected via USB.

A Logitech QuickCam Fusion 1.3 MP Webcam (Logitech, Lausanne, Switzerland) was placed 241 cm off the floor, resting on top of the screen and tilted downwards, approximately 60° from horizontal. This camera was angled so as to capture the participants face and upper body. The webcam recorded videos of participants making their preference choices, with recording starting at the onset of each trial and ending when participants made their response.

Eye movements were recorded using the SMI eye tracking glasses (Senso Motoric Instruments, Teltow, Germany), which feature a scene camera mounted in the bridge of a pair of goggles, as well as binocular eye cameras. The scene camera records from the point of view of the participant, capturing 1280 × 960 video at a rate of 24 fps and with a visual field of 60° horizontally and 46° vertically. Eye position was recorded at 30 Hz, with a manufacturer-estimated spatial resolution of 0.1°.

In the second part of the experiment (the "guess" block), videos of the previous participant from the webcam were replayed on an iMac with a 20-inch monitor (Apple Inc., Cupertino, CA, USA). These videos were rendered at 640 × 480 pixels and presented on a black background at the original speed, playing at 25 fps. Guess responses were entered using the same gamepad as in the preference block.

(a) Preference task (Truth)

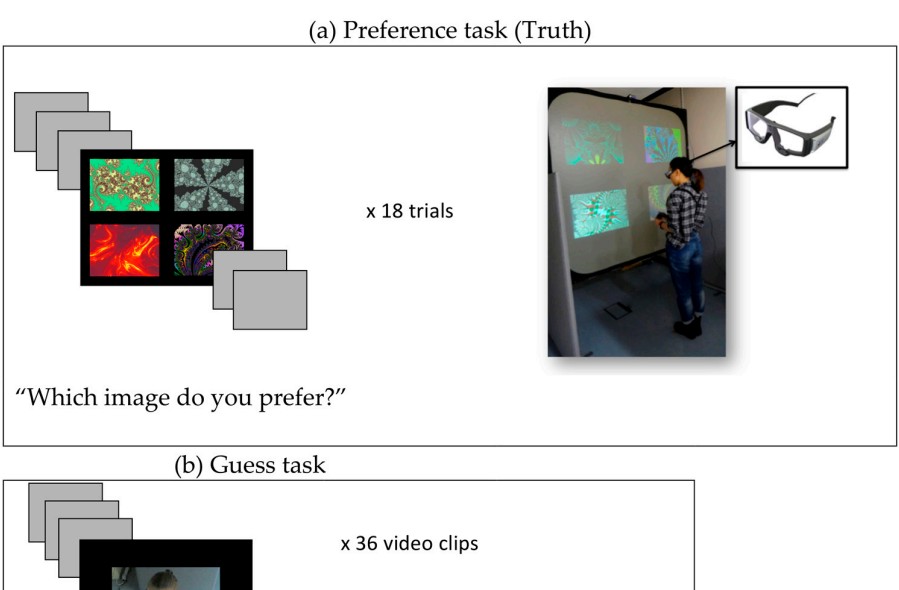

"Which image do you prefer?"

(b) Guess task

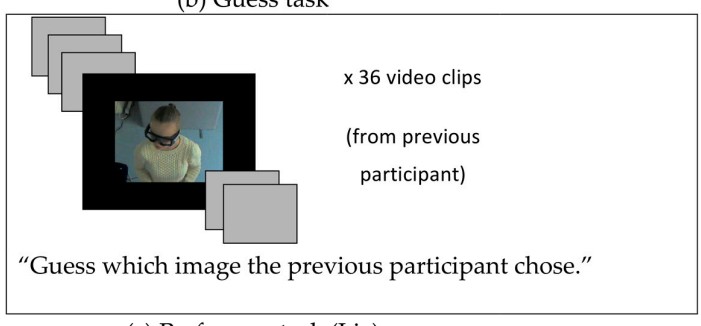

"Guess which image the previous participant chose."

(c) Preference task (Lie)

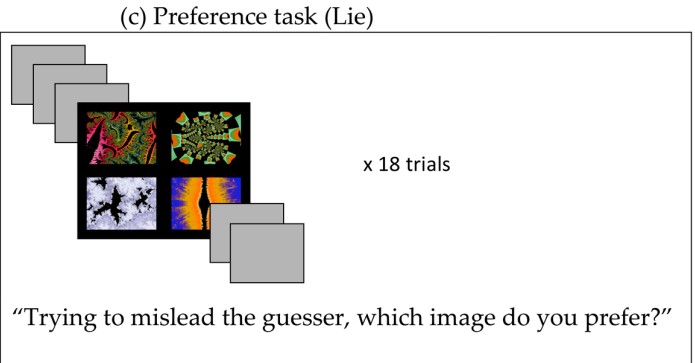

"Trying to mislead the guesser, which image do you prefer?"

**Figure 1.** The procedure in Experiment 1, which consisted of 3 blocks: (**a**) A preference task where participants stood in front of a large screen and chose which of 4 images were preferred, while eye and head movements were recorded; (**b**) An inference task, where videos of the previous observer were displayed and participants had to guess their response; (**c**) A second preference block where participants made a preference judgement while being aware of the guesser and in an attempt to mislead them. A brief version of the instructions is included in each case. Re-played trials in (**b**) came from parts (**a,c**) in the previous participant.

**Procedure and design.** The experiment consisted of three blocks (see Figure 1). In the first and the third block, participants performed a preference task, which required them to choose which of four images they preferred. A "Guess" task was performed in the intervening block, which required participants to guess which picture the previous participant had chosen. All participants completed all three blocks in the same order. Participants gave their consent at the beginning of the study, at which point they were told that their behavior would be recorded via cameras or eyetrackers.

At the beginning of each preference block, a calibration of the eye tracking glasses was performed using SMI's iView software (Senso Motoric Instruments, Teltow, Germany), by asking participants to look at each of five points presented on the screen. The calibration was repeated at the end of the task to make sure that the calibration had not dropped significantly during the task. After

calibration, each trial proceeded with four pictures being presented on the screen. Participants were instructed to choose which of the four images they preferred, and the images remained on the screen until they made their choice by pressing one of the four buttons. The stimulus computer also controlled the webcam, capturing video of the participant in each trial, for the period in which the images were on the screen. In each preference block, 18 randomly assigned trials were completed while eye movements of the participant were recorded. A 1-s blank screen appeared between each trial.

In the first block, no further instructions were given and participants were told only that their eye movements and behavior were being recorded but they should try to forget about that and carry out the task naturally. In the third block, the preference task was repeated and participants were asked to imagine that someone was watching them and to try to mislead the observers. They were instructed that it was important to respond genuinely by choosing their preferred image, but that another person would be trying to guess their choice by watching the webcam. To highlight the key difference between the first and third blocks, we refer to them as the "Truth" and "Lie" blocks, respectively [17].

In the second part of the study, the Guess block, participants were seated in front of a monitor and shown webcam videos of the previous participant. For each video, participants were asked to guess which picture the previous observer had chosen by pressing one of four buttons. An initial practice trial was conducted with the help of the experimenter to ensure that participants understood the mapping between the buttons and the items on the screen. Guesses were made with no assistance or as the best guess if respondents were unsure. Since some of the videos were quite brief, participants had an option to replay the videos by pressing a key on the keyboard. The Guess block consisted of 36 trials which were taken from the Truth and Lie trials of the previous participant. Truth and Lie trials were randomly interleaved.

Importantly, the design and instructions of the experiment meant that participants completed the Guess block immediately after having performed the (Truth) preference task themselves. This ensured that they understood the task, and participants reported finding it easy to make educated guesses about behavior from the webcams. However, observers in the Truth block did not yet know that their own behavior would be classified by other participants, and until the Lie block no-one knew that both honest and deceptive trials were involved in the experiment. Thus, the procedure probed both natural and deceptive orienting, and reading of other participants, despite the fact that each participant was tested alone. At the end of the experiment, participants were debriefed and gave their consent for their videos to be replayed to the next participant.

## 3. Results

We began by examining the guessing performance for Truth and Lie trials, before looking at the orienting behavior displayed in each block.

### 3.1. Guessing Performance

In the guess block, participants made judgements about which of the four images was chosen by the previous observer in each trial. These judgements were made in response to the third-person videos of the observer's head and upper body, and they can be compared to the actual choice made in that trial. Videos were 7.8 s long per trial, on average (SD = 1.9). Informal observation of the participants indicated that they rarely took the opportunity to replay the clips, tending to make their guess after just one viewing. We compared guessing accuracy in Truth and Lie trials, looking at the proportion of correctly guessed trials in each case. Since there were four possible responses, a chance level of accuracy was 25%. The results are shown in Figure 2.

There was a statistically significant difference between guessing accuracy in the Truth and Lie conditions, $t(19) = 3.6$, $p = 0.002$, $dz = 0.80$. We also compared each condition to 25% directly, using a one-sample $t$-test. While accuracy was not high, participants were reliably above chance when drawing inferences from Truth trials, $t(19) = 4.4$, $p < 0.001$, $d = 0.98$. Participants responded at chance levels in Lie trials, $t(19) < 1$.

These results demonstrate that even from a brief and impoverished playback of another participant, naïve, untrained observers can interpret where someone was paying attention and use this knowledge to predict preference—at least some of the time. The most accurate guesser in the Truth block achieved an accuracy of 83%, while the lowest achieved 11%. Importantly, performance in most participants was far worse in the Lie block (M = 21%, range = 0–100%). Thus, the depicted participants were able to change their behavior on these trials in order to mislead the guesser. There was no correlation between performance in the two types of trial (r = 0.06). This is perhaps not surprising, given that participants at this point did not know about the Lie trials and so would have been applying the same strategy throughout. It does however indicate that individual differences in detection in Truth trials did not help or hinder in the Lie trials.

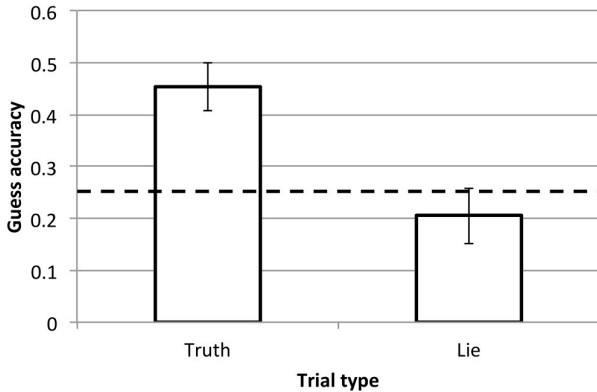

**Figure 2.** Accuracy from the guess block in Experiment 1. Bars show the mean, with standard error bars, for trials from each block. The dashed line indicates chance performance.

Participants made errors about half of the time when guessing in Truth trials. By looking at the pattern of errors we can determine whether participants tended to correctly guess the upper/lower half of the screen or the left/right half of the screen. This would indicate sensitivity to general head direction even though the incorrect item was guessed. Limitations in this sensitivity (e.g., due to the camera angle) could be a source of errors rather than any inability to read behavior in the context of the task. In Truth trials, participants were slightly more likely to guess the correct column (left or right; M = 45% of error trials) than the correct row (top or bottom; M = 35% of error trials), but these proportions were not significantly different (paired t test: $t(19)$ = 1.0, $p$ = 0.31, $dz$ = 0.23). Errors were distributed and not only due to poor sensitivity to horizontal or vertical head position. When responses were collapsed across either left and right or top and bottom, average accuracy was approximately 65% in Truth trials and 40% in Lie trials. The conditions remained different and guessing was only above chance in Truth trials.

### 3.2. Head And Eye Movements

Previous research has established that observers tend to look more at items which they prefer and which they select in situations such as the current task [20]. Foulsham and Lock found that observers were able to mislead by looking at a different item or distributing their fixations in a more random fashion [17]. Analyzing head-free, mobile eyetracking data remains a difficult problem methodologically, since quantifying where on the screen participants were looking would require exact information about head position and field of view. Instead, here we focus simply on the overall number of eye and head movements made during the preference task. Figure 3 shows examples of the raw data in each case. In order to compare across trials of different lengths, we divided the number of head movements and eye fixations in each trial by the duration in seconds (i.e., we used the *rate* of each type of movement). Due to calibration problems and persistent data loss in more than 50% of trials, the data from 2 participants were excluded for these analyses.

Head movements were coded manually from the third-person videos recorded in each trial. A trained research assistant counted the number of large head movements made and coding was completed blind to the condition of the trial or the aim of the experiment. Head movements were counted when separated by a pause where the head didn't move. In practice, most head movements were clear changes in orientation, equivalent to shifts to a different quadrant of the screen. Since this measurement relied on manual coding, a subset of videos was re-coded by a second research assistant as a check for inter-rater agreement. There was an almost perfect correlation between the two raters' estimates of head movement frequency ($r = 0.99$).

There was a significant difference between the rate of head movements in the two conditions, $t(17) = 2.45$, $p = 0.025$, $dz = 0.58$. Fewer head movements were made per second in the Lie trials than in the Truth trials.

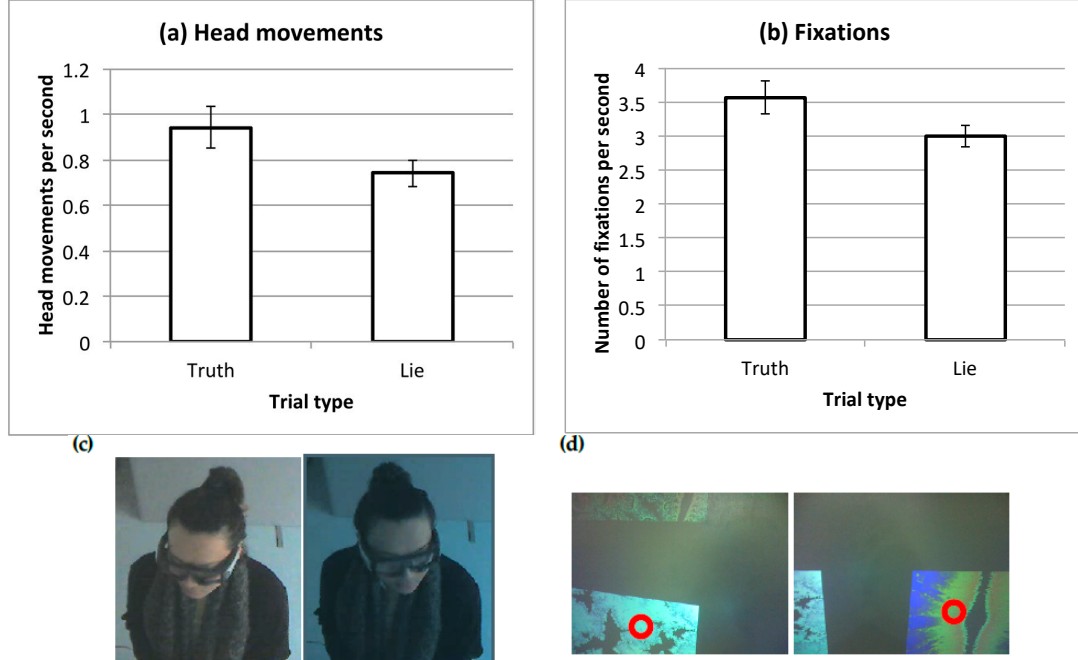

**Figure 3.** Frequency of orienting movements in the two types of trial. Bars show the mean across participants with standard error bars, for (**a**) head movements coded from video and (**b**) fixations recorded by the mobile eyetracker. Examples of the two behaviors are also shown: in (**c**), video frames showing a movement from the observer's right to their left; in (**d**), fixations shown by a circular cursor in the participant's scene camera.

Fixations were parsed using SMI's BeGaze software (Senso Motoric Instruments, Teltow, Germany), which uses velocity thresholds to perform event detection, and which is designed to perform well with head movements. Participants made around 3 fixations per second, which is similar to the fixation rate observed in many laboratory tasks involving images [21–23]. Fewer eye fixations were made in the Lie block than in the Truth block, $t(19) = 2.15$, $p = 0.046$, $dz = 0.51$. This effect was not as large as that with head movements and was observed in 11 of the 18 participants.

## 4. Discussion

The results from this experiment illustrate that participants can infer the attentional state of another person at an above-chance level of performance. In many ways the stimulus presented to guessing participants was less than ideal. Videos were short in duration and low resolution. Guessers tended not to replay them several times, and some of the face may have been obscured by the eyetracking glasses. Guessers did not see the images that were being chosen in each particular trial,

and indeed had to perform some mental rotation to take the perspective of the person in the video. Despite these issues, performance was above chance.

Importantly, introducing the second block of Lie trials changed the socio-communicative context by asking participants to mask their attention and preference. While there could be a number of different ways to do this, participants spontaneously changed their orienting behavior, making less frequent fixations and fewer head movements. These changes did indeed reduce the ability of a third party to read their attention from the video. The effect of concealment on head movements was larger than on eye movements. This might indicate that following the head was more important than following the eyes in this task, which is perhaps to be expected given that head movements are more easily observed. However, detecting these signals may have been hampered by the use of an eyetracker which partially obscured the eye region.

## 5. Experiment 2

In order to test the effect of the presence of eye gaze cues more directly, in Experiment 2, rather than measuring eye movements, we selectively masked some trials by asking the observers to wear sunglasses half of the time. In the no sunglasses condition, the eyes were not obscured by eyetracking glasses, so we might expect this condition to be easier for guessers to follow than in Experiment 1. In the sunglasses condition, any eye gaze clues would be invisible, and participants would be aware that they could move their eyes without being observed. As previously, we examined both how well guessing participants could infer preferences and how head movements changed in the different conditions. This enabled us to examine both the sending of gaze signals and the receiving and interpretation of these signals. If eye movements are important for third-party judgements in this case, then we would expect guessing performance to be worse when the observer is wearing sunglasses. If head movements are sufficient then guessing performance should be equally good when the eye region is masked. In addition, it is possible that wearing sunglasses might change orienting behavior, particularly in the Lie condition. For example, if participants know that their eye signals are shielded, they might prefer to make eye movements and be more likely to reduce their head movements.

*Materials And Methods*

**Participants.** Twenty-eight new student volunteers took part in the experiment (19 females; age range 18–23). Within-subjects effect sizes (*dz*) for the difference between Truth and Lie trials in Experiment 1 were moderate (0.80 for guessing accuracy; 0.58 for head movements). Power analysis indicated that 26 participants were required to have 80% power to detect the smaller of these two effects. As in Experiment 1, an additional pilot participant provided the data for the first participant's guess block.

**Stimuli and apparatus.** The apparatus used was exactly the same as in Experiment 1, with a series of trials displaying 4 images at a time on a large projection screen. However, in this experiment the SMI glasses were not worn and we relied on the webcam recording of the respondent's upper body. In half of the trials, participants were asked to wear a pair of large plastic sunglasses which masked the eyes. In pilot testing we established that these glasses did not impede performance of the task and that the stimuli were still clearly visible.

Since the choice/guess paradigm used here does not require any particular stimuli, this experiment used a different set of images to those in Experiment 1. Each participant responded to sets of 4 photographs depicting frontal views of faces. One hundred and sixty unique female faces were used, sourced from a large online database [24]. The depicted individuals were 18–60 years of age and included both white and black females, facing the camera and with a neutral expression. Stimuli were randomly selected (without replacement), we had no further hypotheses about the preferences for particular faces, and no attractiveness ratings were available. These stimuli were hoped to provide a task which participants would feel more strongly about than rating meaningless fractals, leading to more considered judgements.

**Procedure and design.** The procedure and design were the same as in Experiment 1, with the following exceptions. First, a slightly larger number of trials was used in each block, with 20 sets of faces being viewed in each of the preference blocks. Each participant therefore viewed 40 videos in the guess block, and as previously these replayed all the trials from Truth and Lie conditions of the previous participant. Second, in half of the trials in each block the choosing participant wore sunglasses. This manipulation was blocked and counterbalanced such that participants in the Truth/Lie blocks completed 10 trials with sunglasses followed by 10 trials without sunglasses, or vice versa. During the guess block Truth/Lie trials and sunglasses/no sunglasses trials were randomly interleaved.

## 6. Results

### 6.1. Guessing Performance

The videos replayed in the guess block in this experiment were 6.9 s long, on average (SD = 2.3). Figure 4 shows the guessing accuracy from the second part of the task, averaged within Truth and Lie trials and those where the target in the video wore or did not wear sunglasses. Performance in this condition was slightly better than in Experiment 1, and well above chance in the Truth condition. However, the wearing of sunglasses had absolutely no influence on guessing accuracy. A 2 × 2 repeated measures ANOVA confirmed that there was a significant main effect of block, with better guessing accuracy in the Truth block than in the Lie block, $F(1,27) = 35.482$, $p < 0.001$, $\eta_p^2 = 0.57$. There was no main effect of sunglasses and no interaction (both $F(1,27) < 1$). One-sample $t$-tests confirmed that both conditions were above chance in the Truth block ($t(27) > 7$, $p < 0.001$, $d > 1.3$). In the Lie block, neither condition was significantly different from chance ($t(27) \leq 1.0$). There were, however, large individual differences, with accuracy ranging from 10% to 100% in Truth and from 0% to 100% in Lie.

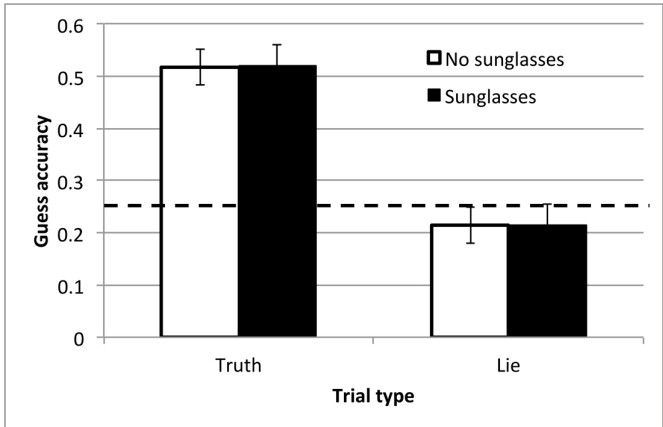

**Figure 4.** Accuracy from the guess block in Experiment 2. Bars show the mean, with standard error bars, for trials from each block. The dashed line indicates chance performance.

As in Experiment 1, we also checked whether people were mostly able to guess the row (top or bottom) or column (left or right), even when they made an incorrect guess in the Truth trials. Participants were again more sensitive to horizontal changes in position, and in this experiment,    this was a significant difference. Of the erroneous responses, 46% correctly guessed whether the chosen image was on the left or the right, whereas only 26% correctly guessed whether it was on the top or the bottom (paired $t$-test: $t(27) = 3.7$, $p = 0.001$, $dz = 0.69$). Collapsing the responses into just left and right gave a mean accuracy of 73% in Truth trials. This remained much higher than in Lie trials (41%) and did not change the null effect of sunglasses.

### 6.2. Head Movements

As in Experiment 1, we manually coded the number of head movements in each trial and then standardized this by the duration of the trial. Due to recording errors, the videos from 3 participants could not be coded. The results for the remaining participants are shown in Figure 5.

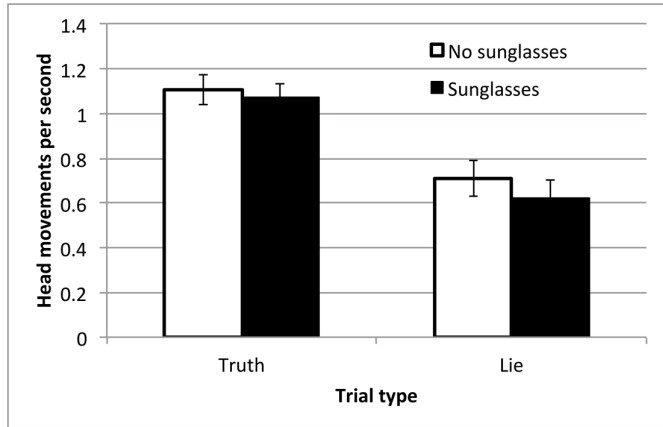

**Figure 5.** Mean rate of head movements, across the conditions in Experiment 2, and with standard error bars.

There was a clear and significant effect of block on the rate of head movements, $F(1,24) = 27.8$, $p < 0.001$, $\eta_p^2 = 0.54$. As in Experiment 1, Lie trials were associated with fewer head movements per second, which is consistent with the idea that participants are trying to mislead by masking the most obvious cues to their attention. There were slightly fewer head movements made when the observer wore sunglasses, but this effect did not reach significance, $F(1,24) = 3.07$, $p = 0.09$, $\eta_p^2 = 0.11$. There was no interaction between block and sunglasses, $F(1,24) < 1$.

## 7. General Discussion

The results from Experiments 1 and 2 show clearly that third-party observers can predict natural performance in a preference task by observing overt attention. Although the observer videos were in some ways impoverished, and participants could not see the stimuli being responded to, they contained enough information to make guesses with accuracy above chance. This concurs with other research suggesting that third-parties can show important insights into attention in different settings [25]. A likely explanation in the present study is that participants tend to gaze more at items that they prefer or will subsequently choose [20]. In Foulsham and Lock, participants also tended to fixate the chosen item last [17]. Observers in the guess block here were presumably able to make a good estimate of gaze direction and, in the context of the particular task, use this gaze direction to make an inference about judgement.

While performance in the two experiments reported here was firmly above chance, participants still only got approximately 50% of trials correct, a worse level of accuracy than the approximately 70% in a previous study with accurate fixation information [17]. This difference is likely to be because extracting gaze information was more difficult from the videos. Guessers made errors on about half of the trials. It appears that at least some of these errors were because it was hard to precisely differentiate between gazes on different quadrants. This is demonstrated by looking at the distribution of errors in each of the four quadrants, which shows that participants were likely to guess the correct side of the screen (top/bottom or left/right), even when they didn't get the exact quadrant. This was true in both experiments, and in fact the horizontal position was guessed more often than the vertical position. This is probably because the stimuli were wider than they were high (making horizontal gaze movements larger) and because the camera angle made it harder to perceive up and down head movements. This could be tested with different camera angles or a higher resolution video.

In the task used here, observers in the guess block could potentially have used both eye direction and head direction, as well as postural cues, in order to infer the focus of attention. However, the results from Experiment 2 demonstrate clearly that head movements are more important, since performance was unaffected when the eyes were masked by sunglasses. Of course, it is well established that, when required by the task, participants are able to detect small changes in eye position and follow these gaze signals [26]. It is likely that if our task and stimuli required noticing smaller, fine-grained differences in spatial attention, and if the eyes were easier to detect in our videos, we would have found more of an effect of masking with sunglasses. Nonetheless, this result emphasizes that head movements are a clear measurable marker of orienting in complex situations.

Importantly, the present experiments also show that the signaling function of gaze [2,15], can be modified in a flexible manner according to the social context. When required to deceive someone who is watching, participants in both experiments reduced the frequency with which they made head movements. Presumably this is because these participants knew that head movements can be an easier signal for observers to pick up on. Moreover, this change in orienting behavior proved successful, because participants attempting to guess based on Lie trials were unable to do so better than chance. Interestingly, there are a number of reports investigating lie detection which suggest that liars, in general, make more head movements than those telling the truth [27,28]. However, in the present study, where head movements were given a clear spatial and signaling function, participants were able to reduce these signals.

One of the differences between the two experiments presented here was in the stimuli that observers were asked to choose in the preference task. In Experiment 1, participants selected among colorful fractal images. In Experiment 2, they chose one of 4 photographs of female faces. On the one hand, there is good reason to think that both variations will elicit similar choice behavior. Previous reports of gaze biases have shown similar behavior (e.g., looking more at what you are going to choose in a preference task) in a number of different categories including faces, fractals, shapes and scenes [17,20,29]. The present results also show that, at least for the task of inferring attention used in the guess block, gaze shifts for both fractals and faces are both interpretable. Informal inspection of videos from the two experiments suggests that orienting behavior was not markedly different in the two experiments (and the rate of head movements was similar). On the other hand, as described in the introduction, faces are often attended to in a different way, at least when they are in competition with other stimuli. To the extent that the preference task in Experiment 2 was a social judgement, participants may have behaved differently from the non-social task in Experiment 1. A more detailed analysis of how overt and covert attention is deployed when looking at faces would be worthwhile, and it is likely that behavior would be very different with real faces than with images of faces (the task used here). Nevertheless, our results show that in both experiments orienting can be interpreted by a third party, and head movements can be changed in order to mislead. Guessing performance in the Truth condition was slightly better in Experiment 2. Further research would be necessary to determine whether this is reliable and whether it is due to changes in guesser motivation or actual differences in signaling with different stimuli.

The present study provides an example of how a laboratory paradigm can study attention during both sides of an interaction. Although there was not another person physically present and trying to guess the observer's choices, the task was designed so that, in the Lie block, participants were reacting to the social context implied by the webcam. There remain a number of interesting questions about how participants might balance attentional orienting mechanisms in this task. For example, since participants were moving their head (and eyes in Experiment 1) less frequently in the Lie block, does this mean that they were paying less attention to their preferred item? Or were they using covert attention more often? There was also weak evidence that masking one attentional signal (by wearing sunglasses) led to fewer head movements. Such results indicate that, although a large amount of research uses eye movements to quantify attention, overt orienting may involve a choice of effectors. We propose that levels of attention–head movements, eye movements and covert selection—can be flexibly coordinated according to the social context.

**Author Contributions:** Conceptualization, T.F.; Investigation, M.G. and L.C.; writing—original draft preparation, T.F., M.G. and L.C.; writing—review and editing, T.F.

**Funding:** This research received no external funding.

**Conflicts of Interest:** The authors declare no conflict of interest.

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
