# Peer review of "Reading and Misleading: Changes in Head and Eye Movements Reveal Attentional Orienting in a Social Context"

_2411-5150, 2019_

Round 1

Reviewer 1 Report

Review of ‘Reading and Misleading: Changes in Head and Eye Movements Reveal Attentional Orienting in a Social Context’ for Vision

Summary

The authors present two experiments that assess the extent to which we are able to guess the preference of other people based on exposure to their looking behaviours.

Both Experiments contained 3 blocks:

Block one had participants look at scenes containing 4 pictures and their looking behaviour was recorded via camera. On each trial their task was to indicate via a controller which of the 4 quadrants they preferred.

Block 2 had participants watch the recorded looking behaviour of the prior participant (blocks 1 and 3, interleaved). After each trial (a clip showing the looking behaviour before each choice judgement), participants had to guess which of the 4 quadrants the person in the video had preferred.

Block 3 was the same as block 1, except that participants were instructed that an observer would try to guess their preference, so they should try to conceal their preference.

Experiment 2 had the additional variable of whether the looker’s eyes were visible, or occluded by sunglasses.

The findings generally indicate that:

-          An observer is above chance at ascertaining the favoured quadrant based on the gaze behaviour observed when the looker was under no instructions.

-          An observer is at chance level performance at ascertaining the favoured quadrant based on the gaze behaviour observed when the looker was told to ‘lie’.

-          When ‘lying’ people make less head and eye movements.

-          Sunglasses make no difference

Thoughts

I generally like the paper. I do however have one key point that I feel must be addressed prior to publication and offer 4 other smaller suggestions below.

Key point:

The authors report 50% performance on ‘truth’ trails. While this is higher than chance level of 25% (choosing out of 4 options), it is concerning to me that this 50% performance could reflect that observers are only able to guess the gross orientation of social orienting (e.g. up vs down, left vs right).

If this was the case, the data are still informative, but perhaps the conclusions would need to be a little different – it would mean that people were guessing more than the authors currently assume.  

For example, if the correct choice on a given trial was the top left, are all of the incorrect choices (50% of the data) consistently bottom left, top right, or a mixture of each? This might shed light on whether observers really are working out which quadrant was preferred, as opposed to only being able to vaguely work out where on the screen (which half) was looked at.

Other points:

Mobile eye-tracking is tricky, so I get while a fine grained analysis was not conducted. But, the rationale for choosing number of head movements/ fixations was not clear to me (from the intro). Should we expect less social orienting when trying to conceal our thoughts? Or might we predict equal amounts of orienting, but orienting to other parts of the display?  

I don’t like the use of ‘marginally significant’ for a p=.09 (Exp 2), more appropriate to acknowledge the pattern of the data and let readers attribute the importance of this patterns given the stats.

Exp 1 description of ‘reliably above/ below chance’ should be reconsidered – the t-test is not comparing to chance (e.g. the performance of ‘lie’ trials could surely just as appropriately be described as ‘at chance level’?)

This sentence (Exp 1 discussion) needs reconsideration: “The difference between the two measures may 254 indicate that the head was more important than the eyes in this task, which is perhaps to be expected 255 given that head movements are more easily observed” – I think the authors mean that the data was more clear (larger effect) for head movements compared to fixations, but this should be made clear.

Reviewer 2 Report

The paper is well-written, and presents a cleverly-designed study. While some of the points below are more minor, there are a few key points that may require more work to address.

Points

1) While there is evidence that “people are special and may trigger automatic capture of visual attention”, there is also evidence to the contrary, for which the authors should make note. For example, Pereira, Birmingham & Ristic (2019, Psychological Research) found no biasing of attention towards faces. There are also other studies indicating lower levels of attention capture by social faces in everyday situations (e.g., Gallup et al., 2012a, Biology Letters; 2012b, PNAS), however if the author’s first point is specific to lab-based tasks, I would suggest making this more explicit.  

2) The authors should include additional citations demonstrating changes of attention, cognition, and behaviour when in the presence of others, such as the investigations by Gallup et al. (2012a, Biology Letters; 2012b, PNAS).

3) Lines 39-41: “When these two sets of findings are considered together, it is perhaps not surprising that a number of recent studies have found discrepancies between the way that participants respond to images of people and the way that they respond to people who are physically present [2].” While the Risko et al. paper cited is a review, and thus emcompasses more than one study, the authors should consider adding at least one extra reference, to match their wording of “a number of recent studies”. For example, Hayward et al. (2017; Canadian Journal of Experimental Psychology) found this very discrepancy within the same participants using both computerized and real-world measures.

4) Can the authors provide justification (via sample size/effect size calculations) for why they chose 20 participants for E1 and 28 for E2?

5) I appreciate the clever design, with having the “Truth” task and “Guess” tasks before the “Lie” task, in order to preserve the integrity of each of the phases of the experiment. I would like some clarification on a few points about the setup, however. First, how did participants provide consent to be videotaped, and what was the wording/rationale for this aspect of the methods? Second, did the authors collect any information (subjective or otherwise) as to whether participants noticed in the “Guess” trials that some were easier to discern than others (and if so, were participants good at classifying the two types of trials)?

6) Lines 185-186: “Participants took an average of about 3.5s per trial to make their guess, which indicates that they rarely took the opportunity to replay the clips.” Can the authors please provide information as to how long the clips were, on average?

7) Lines 198-199: “There was no correlation between performance in the two types of trial (r = .06), perhaps indicating that the strategy used for inference in Truth trials neither helped nor hindered in the Lie trials.” Why would the authors expect the strategy to change between Truth and Lie trials? As the two types of trials were intermixed and participants had not yet been made aware of the “Lie” condition, presumably the participants would be employing the same strategy across all trials.

8) The authors provide information as to the most accurate guesser in the “Truth” condition (line 196). Can they also provide information as to the least accurate guesser, along with those values for the “Lie” condition? One informative way that the authors could present the individual participant data would be to use a stripchart, this way the readers have a sense of performance distribution.

9) Figure 3(b): on my copy, the figure contains a number of “?” symbols (see also the references).

10) How good were participants at judging which side of the screen the preferred stimulus was located? From the video angle, a left-right judgement is presumably easier to judge than top versus bottom. After collapsing across top and bottom, are there still differences between the “Truth” and “Lie” conditions?

11) Lines 235-237: “Participants made around 3 fixations per second, which is similar to the fixation rate observed in many laboratory tasks involving images [16].”. The authors should provide at least one other reference for this statement, or revise the statement to remove “many laboratory tasks”.

12) In E2, why were only female faces chosen as the stimuli? What are the demographics of the face stimuli (e.g., age range, race, etc)? Were the more attractive faces more likely to be chosen as the preferred stimulus?

13) Do the authors believe that asking participants to make a ‘social’ judgement (in E2) is no different than making a ‘nonsocial’ judgement (in E1)? I know the authors allude to their rationale for choosing faces in E2 on lines 285-287 (“These stimuli were hoped to provide a task which participants would feel more strongly about than rating meaningless fractals, leading to more considered judgements.”), however their statement does not get at the social/nonsocial distinction. If attention to faces/gaze is ‘special’, then perhaps there is something different about how we overtly and covertly attend to faces, fundamentally changing how participants do the “Guess” task, and also makes it more difficult to directly compare the findings across the two experiments. At minimum, the authors should speak to this point in the manuscript, and potentially collect additional data to speak to this important point.

14) Do the authors believe that the improvements seen in the E2 “Truth” condition (as compared to the E1 “Truth” condition) is due to using different stimuli?

15) Why did the authors choose not to analyze the number of fixations in E2? I suggest including this analysis, even if non-significant, in order to stay constant with the analyses presented in E1.

16) Lines 353-354: “Importantly, the present experiments also show that the signaling function of gaze can be modified in a flexible manner according to the social context.” The authors should cite Gobel et al. (2015; Cognition) and/or Risko et al. (2016; Current Directions in Psychological Science), who both discuss the importance of investigating the signaling function of gaze.

Round 2

Reviewer 1 Report

The authors have done a great job in dealing with the suggestions that I raised in my initial review and I am happy to recommend publication. 

Reviewer 2 Report

The authors have done a good job of addressing all of my points. I have no other concerns, and look forward to seeing this manuscript published.